# Declining Prevalence of SARS-CoV-2 Antibodies among Vaccinated Nursing Home Residents and Staff Six Months after the Primary BNT162b2 Vaccination Campaign in Belgium: A Prospective Cohort Study

**DOI:** 10.3390/v14112361

**Published:** 2022-10-26

**Authors:** Eline Meyers, Ellen Deschepper, Els Duysburgh, Liselore De Rop, Tine De Burghgraeve, Pauline Van Ngoc, Marina Digregorio, Simon Delogne, Anja Coen, Nele De Clercq, Laëtitia Buret, Samuel Coenen, An De Sutter, Beatrice Scholtes, Jan Y Verbakel, Piet Cools, Stefan Heytens

**Affiliations:** 1Department of Diagnostic Sciences, Faculty of Medicine and Health Sciences, Ghent University, 9000 Ghent, Belgium; 2Biostatistics Unit, Faculty of Medicine and Health Sciences, Ghent University, 9000 Ghent, Belgium; 3Department of Epidemiology and Public Health, Sciensano, 1000 Brussels, Belgium; 4EPI-Centre, Department of Public Health and Primary Care, KU Leuven, 3000 Leuven, Belgium; 5Research Unit of Primary Care and Health, Department of General Medicine, Faculty of Medicine, University of Liège, 4000 Liège, Belgium; 6Department of Public Health and Primary Care, Faculty of Medicine and Health Sciences, Ghent University, 9000 Ghent, Belgium; 7Department of Family Medicine & Population Health, Faculty of Medicine and Health Sciences, University of Antwerp, 2000 Antwerp, Belgium; 8Nuffield Department of Primary Care Health Sciences, University of Oxford, Oxford OX1 2JD, UK

**Keywords:** seroprevalence, SARS-CoV-2, IgG, IgM, nursing home residents, elderly, COVID-19 vaccination, BNT126b2

## Abstract

In the SCOPE study, we monitored SARS-CoV-2 antibodies in a national sample of residents and staff from Belgian nursing homes. Here, we report the seroprevalence among infected and infection-naive residents and staff after the primary COVID-19 vaccination campaign. Among 1554 vaccinated nursing home residents and 1082 vaccinated staff from 69 nursing homes in Belgium, we assessed the proportion having SARS-CoV-2 antibodies approximately two (April 2021), four (June 2021), and six months (August 2021) after a two-dose regimen of the BNT162b2 vaccine. We measured the seroprevalence using SARS-CoV-2 antibody rapid tests and collected socio-demographic and COVID-19 medical data using an online questionnaire. Two months after vaccination (baseline), we found a seroprevalence of 91% (95% CI: 89–93) among vaccinated residents and 99% (95% CI: 98–99) among vaccinated staff. Six months after vaccination, the seroprevalence significantly decreased to 68% (95% CI: 64–72) among residents and to 89% (95% CI; 86–91) among staff (*p* < 0.001). The seroprevalence was more likely to decrease among infection-naive residents, older residents, or residents with a high care dependency level. These findings emphasize the need for close monitoring of nursing home residents, as a substantial part of this population fails to mount a persistent antibody response after BNT162b2 vaccination.

## 1. Introduction

During the severe acute respiratory coronavirus 2 (SARS-CoV-2) pandemic, nursing home (NH) residents (NHR) have been disproportionately affected by COVID-19 [1]. Therefore, in Belgium, from January to March 2021, NHR and nursing home staff (NHS) were the first to receive a two-dose regimen of the BNT162b2 (Pfizer-BioNTech) vaccine [2]. Although the phase I/II and III trial data of this vaccine showed effective immunization and robust clinical protection, follow-up data were limited to two months after complete vaccination at the moment of market authorization [3,4]. Moreover, these trials only assessed the immune responses upon vaccination among persons under 85 years of age [3,4], while the average age of an NHR in Belgium is 87 years of age [5]. Additionally, NHR are known to suffer from multimorbidity [6]. Both older age and comorbidity have been previously associated with impaired humoral immune responses, one of the hallmarks of immunosenescence [7,8,9,10]. Recent studies in the general population have shown waning humoral immune responses within the first six months after a two-dose regimen of BNT162b2 vaccination [11,12,13]. However, data on SARS-CoV-2 immunity upon BNT162b2 vaccination in the age category above 80 years old remain limited.

Therefore, in the SCOPE (SARS-COv-2 seroPrEvalence) study, we longitudinally monitored the prevalence of SARS-CoV-2 antibodies in residents and staff from 69 Belgian nursing homes. Here, we report the seroprevalence among a cohort of 1554 residents and 1082 staff (for comparison) two, four, and six months after the first two doses of the BNT162b2 vaccine.

## 2. Materials and Methods

### 2.1. Study Design and Sample Size

The current publication reports longitudinal data collected within the SCOPE study, which is part of a national sero-epidemiological surveillance program.

The SCOPE study was designed as a national prospective cohort study that bimonthly assessed the prevalence of SARS-CoV-2 antibodies in Belgian NHR and NHS between February and December 2021. We assumed a seroprevalence rate of 0.5 at the start of the study and a half-width of a 95% confidence interval (CI) of 0.05, and a drop-out rate over ten months of 0.2 and 0.4 for NHS and NHR, respectively. As residents/staff are clustered within care facilities, the sample size was augmented by the design effect, assuming an intraclass correlation coefficient (ICC) of 0.12. Therefore, we aimed to recruit 24 NHR and 20 NHS in 69 NH in Belgium, bringing the total sample size to 1656 NHR and 1380 NHS.

This publication comprises the SARS-CoV-2 seroprevalence results from April, June, and August 2021 among the cohort that was fully vaccinated at the moment of antibody testing in April 2021. The results of the test rounds in February, October, and December 2021 are not included in this publication, as in February, the primary vaccination campaign was ongoing, and starting from October 2021, booster vaccines were administered to NHR and NHS.

### 2.2. Study Population and Recruitment

In the SCOPE study, we used a two-stage cluster sampling to obtain a random sample of 1656 NHR and 1380 NHS among 69 NHs, in order to recruit the total cohort. In the first stage, 69 NH were selected out of a total of 1521 Belgian NH. In order to recruit a sample of NH that was evenly spread across the Belgian territory, the selection of NH was randomly sampled within strata defined by regions and provinces proportionally to the population. When a NH refused to take part in the study, the first NH in the spare list for that province was contacted. For the random selection of participants, every participating NH made an alphabetic list of all eligible NHR (service flat residents were excluded) and NHS (temporary NHS employed for a period < 1 year after baseline collection, or NHS < 18 years old, were excluded). From this list, 24 NHR and 20 NHS were randomly selected and invited to participate. When a participant refused to participate, the first following participant from a spare list was invited.

### 2.3. Ethical Considerations

The SCOPE study was approved by the Ethics Committee of the Ghent University Hospital (reference number BC-08719) and conducted according to the principles outlined in the Declaration of Helsinki. Each participant or their legal representative signed an informed consent form after being informed about the goal of the study and the study procedures.

### 2.4. Data Collection

#### 2.4.1. Antibody Testing

We assessed the presence of SARS-CoV-2 antibodies in capillary blood collected by a finger prick using point-of-care COVID-19 IgG/IgM rapid test cassettes (Healgen Scientific LLC, Houston, TX, USA). These rapid tests have reported sensitivity of 92.9% and a specificity of 96.3% [14]. Testing was performed by trained study staff; however, if a participant was absent during the NH testing visit, a detailed instruction sheet was left to have sampling performed by NH medical staff. Sample collection in April 2021 started in a staggered way, on 31 March 2021, so all samples among the different NH were collected within the first ± four weeks after collection of the first sample. Follow-up samples in June and August 2021 were collected 60 ± 7 and 120 ± 7 days after the NH collection in April 2021, respectively.

#### 2.4.2. Questionnaire

We asked each participant to complete an online questionnaire (LimeSurvey version 3.22, LimeSurvey GmbH, Hamburg, Germany) on the day of antibody testing. The NH head nurse(s) completed the questionnaires for NHR. Participant characteristics (e.g., age, sex, job type, care dependency level (for NHR)) and relevant COVID-19 comorbidities (cardiovascular disease, diabetes, hypertension, immunosuppression, severe renal/lung/cardiac disease, active cancer) were recorded. Additionally, at every test round, we asked the participants about their COVID-19 vaccination status (number of doses, date, vaccine type by brand name) and infection status (as assessed by previous PCR and/or antigen test and/or CT scan, with the respective test results, and date of testing in case of a positive test result).

### 2.5. Statistical Analysis

We estimated the seroprevalence as the number of participants with a positive IgG and/or IgM test divided by the total number of valid test results. The prevalence of SARS-CoV-2 antibodies was reported stratified for NHR and NHS, with the 95% confidence interval (CI), assessed by a generalized equation estimation (GEE) analysis with independence covariance structure, binomial family and logit link, and robust variance estimator. CIs for seroprevalence were back-transformed from the logit scale. The seroprevalence was estimated overall stratified for NHS and NHR and by time point and age category based on a GEE analysis with exchangeable covariance structure. In addition, to model the evolution of the seroprevalence over time, a stratified analysis for NHS and NHR was performed by a GEE analysis with an unstructured correlation structure for the repeated measurements within each participant from month 2 to month 6. To assess the associations between the detection of SARS-CoV-2 antibodies (seropositivity) and the time-varying history of a SARS-CoV-2 infection (either PCR- and/or antigen test- and/or CT-confirmed; self-reported by participants or reported by nursing staff for NHR), a time x time-varying self-reported SARS-CoV-2 infection interaction was included in the model, and adjusted for baseline age, sex, comorbidities and care dependence level (the latter two for residents only). Adjusted odds ratios for seropositivity were reported with 95% CI. Statistical analyses based on GEE models were performed on participants who had valid rapid test results and completed the survey to derive the vaccination status and the history of a SARS-CoV-2 infection. In case of missing survey data, no change in self-reported history of a SARS-CoV-2 infection was assumed at the missing time point (Last Observation Carried Forward principle). Two-sided *p*-values of ≤0.05 were considered statistically significant. Analyses were performed in R, version 4.0.2, using the GEE-library (version 4.13-20) and emmeans-library (version 1.5.0).

## 3. Results

### 3.1. Participation

Out of the total recruited cohort, 1554 NHR and 1082 NHS were fully vaccinated before April 2021 and included in the current analysis. At baseline (April 2021), a total of 1475 COVID-19 NHR and 1040 COVID-19 NHS from 69 NH across Belgium had a valid SARS-CoV-2 antibody test result. For the follow-up visits in June and August 2021, antibody test results were available for 1408 NHR and 959 NHS and for 1323 NHR and 824 NHS, respectively (Figure 1).

### 3.2. Participant Characteristics

The median age of NHR was 87 years (interquartile range (IQR) 81–91), and the majority was female (75%). Most NHR (68%) suffered from at least one of the following comorbidities: cardiovascular disease (41%), hypertension (34%), diabetes (17%), severe heart-/lung-/renal disease (11%), active cancer (4%), immunosuppression (2%). Among residents, 39% were highly care-dependent (care level C, Cd, and D), while 61% were independent to intermediately dependent (care level O, A, B), as classified by the care-dependency evaluation scale (Katz Index) [15]. The majority of (self-)reported SARS-CoV-2 infections (either PCR- and/or antigen test- and/or CT-confirmed) occurred before vaccination. Before administration of the first dose, 36% of NHR reported a history of SARS-CoV-2 infection. By April and June 2021, 38% of NHR had a history of SARS-CoV-2 infection. By August 2021, 39% of NHR reported a history of SARS-CoV-2 infection.

The median age of NHS was 43 years (IQR 34–53), and the majority was female (84%). Among NHS, 18% suffered from at least one comorbidity: hypertension (11%), cardiovascular disease (3%), diabetes (3%), severe heart-/lung-/renal disease (2%), immunosuppression (2%), active cancer (1%). The majority of (self-)reported SARS-CoV-2 infections among NHS occurred before vaccination. Before administration of the first dose, 29% of NHS reported a history of infection; by April and June, 31%, and by August, 32%.

All vaccinated participants were administered a BNT162b2 vaccine except for one participant (NHS) who received an mRNA-1273 vaccine (Moderna). The mean time between the first and second vaccination was 21 days. Antibody testing at baseline (April 2021) occurred on average 65 and 62 days after administration of the second dose for NHR and NHS, respectively.

### 3.3. Prevalence of SARS-CoV-2 Antibodies among NHR and NHS Four and Six Months after BNT162b2 Vaccination

Table 1 shows the SARS-CoV-2 seroprevalence among NHR and NHS (per age group) two, four and six months after COVID-19 vaccination.

At baseline (April 2021), approximately two months after NHR and NHS received their second dose of the BNT162b2 vaccine, SARS-CoV-2 antibodies were present in 91% (95% CI; 89–93) NHR and 99% (95% CI; 98–99) of NHS. The majority of NHR (96%) who did not have detectable antibodies upon vaccination (non-responders) were infection-naive. Among NHR antibody responders, 58% were infection-naive.

Four months after vaccination (June 2021), the seroprevalence dropped to 87% (95% CI; 83–89) among NHR, while among NHS, it remained stable at 99% (95% CI; 98–100). After six months (August 2021), the seroprevalence among NHR further decreased to 68% (95% CI; 64–72). Among NHS, the seroprevalence also decreased six months after vaccination, with 89% (95% CI; 86–91) of staff having SARS-CoV-2 antibodies. The observed decreases in SARS-CoV-2 seroprevalence were statistically significant (*p* < 0.001).

Overall, the prevalence of SARS-CoV-2 antibodies was lower with older age; however, these differences were only statistically significant for NHR (*p* = 0.001) (Table 1 and Table 2).

Figure 2 shows the SARS-CoV-2 seroprevalence over time among NHR and NHS with and without a self-reported history of COVID-19 infection. A significantly larger decline in seroprevalence was observed among infection-naive NHR within 6 months after vaccination, compared to previously infected NHR (*p* < 0.001, Table 2). Among NHS, no significant differences were observed between the two groups (infection-naive vs. previously infected) (*p* > 0.05, Table 2).

### 3.4. Association between SARS-CoV-2 Seropositivity and Age, Comorbidities, Care Dependency, Self-Reported History of SARS-CoV-2 Infection, and Time after Vaccination in NHR and NHS

The odds of SARS-CoV-2 seropositivity among NHR and NHS are presented in Table 2. Compared to the baseline (April 2021), NHR had significantly lower odds of being SARS-CoV-2 seropositive 4 and 6 months after vaccination (*p* < 0.001). NHS only had significantly lower odds of being SARS-CoV-2 positive 6 months after vaccination (*p* < 0.001) compared to baseline (April 2021).

NHR with a self-reported history of COVID-19 infection had significantly higher odds of being SARS-CoV-2 seropositive (*p* < 0.001), while this was not observed for NHS. On the other hand, NHR with a high care dependency level (level C/Cd/D) or older age (>87 years old) had significantly lower odds of having SARS-CoV-2 antibodies (*p* = 0.002; *p* = 0.001).

## 4. Discussion

In a population of 1554 vaccinated NHR and 1082 vaccinated NHS in Belgium, we tested for the presence of SARS-CoV-2 antibodies approximately two (April 2021), four (June 2021), and six months (August 2021) after BNT162b2 two-dose regimen vaccination. We demonstrated that the prevalence of SARS-CoV-2 antibodies significantly dropped within six months after vaccination, with the largest decrease observed in NHR. NHR who were infection-naive, older NHR, or NHR with a high care dependency level were most likely to become seronegative.

This study demonstrates impaired antibody responses upon COVID-19 vaccination in NHR, a generally frail population of older age, compared to NHS, a generally healthy and working-age population. Likewise, supporting evidence exists on impaired antibody responses in NHR upon BNT162b2 vaccination. A study by Canaday and colleagues showed that antibody concentrations upon vaccination were significantly lower in infection-naive NHR than infection-naive NHS [16]. Moreover, they showed significant decreases in SARS-CoV-2 antibody concentrations 6 months after BNT162b2 vaccination for both NHS and NHR, but with the lowest antibody levels observed among infection-naive NHR. Their study showed that 6 months post-vaccination, 69% of the infection-naive NH residents had neutralization titers at or below the lower limit of detection, supporting our study findings [17]. The influence of old age on BNT162b2 vaccine-induced humoral responses was also demonstrated by Müller and colleagues, who showed a significantly decreased mean antibody titer in the age group above 80 compared to younger vaccinees [18]. Similarly, a large-scale study by Levin and colleagues found that humoral immunity waned six months after vaccination, with lower neutralizing antibody titers among persons 65 years of age or older [11]. Additionally, a study from Witkowski et al. demonstrated that approximately 11% of NHR did not present a humoral immune response after COVID-19 vaccination, which is in line with our findings [19]. They demonstrated that these non-responders also presented a less potent T-cell immunity than NHR responders.

It is clear that NHR, in particular, are prone to impaired and rapidly waning antibody response after COVID-19 vaccination. The underlying cause behind these observations might be explained by immunosenescence, the age-related dysfunction of the immune system. It has been previously described for other vaccines targeting other respiratory tract infections that, likewise our observations, old age, and frailty are strongly associated with decreased immune responses upon introduction of newly encountered antigens [7,8,9,20,21,22].

As other studies have demonstrated that binding and neutralizing antibody responses are strongly correlated with COVID-19 protection, it is suggested that the decreasing seroprevalence observed in the NHR population also might have implications on their clinical protection against COVID-19 [23,24]. This is particularly worrying, as NHR are at high risk for COVID-19 hospitalization and mortality. A study based on data from the general population in the USA has shown that effectiveness against COVID-19 infection declined to 47% 5 months after BNT162b2 vaccination, with the highest decline in effectiveness observed among the 65+ age category. However, effectiveness against hospital admissions remained high for up to six months, which is, together with the prevention of mortality, one of the most important outcomes of vaccination [25]. Similar observations were made among Belgian NHR, as despite the declining seroprevalence, the number of COVID-19 hospitalizations among NHR remained low (≤ 0.08 weekly COVID-19 hospitalizations/100 NHR) in the following 10 months after completion of the vaccination campaign [26]. These data suggest that a decreasing SARS-CoV-2 seroprevalence does not immediately imply a thread for NHR’s protection against COVID-19 hospitalization. However, in October 2021, Belgium started administering booster vaccines to NHR, which possibly also contributed to the low number of COVID-19 hospitalizations observed. Similar studies will be needed to monitor antibody responses upon booster vaccination in NHR to address whether antibody responses remain persistent across time. Especially for NHR, who are characterized by their old age, frailty, and multimorbidity, making them vulnerable to severe cases of COVID-19. Additionally, the association between the prevalence of SARS-CoV-2 antibodies and hospitalization and mortality needs to be further investigated, as currently, it is not known which immunological marker is predictive for COVID-19 protection.

Although important observations were made regarding declining seroprevalence, this study has limitations. Primarily, the sampling periods in this study were independent of the moment of vaccination of participants, meaning that the time between vaccination and antibody testing differed among participants. Baseline sampling occurred in April 2021, while the mass vaccination campaign officially ended on 24 March 2021. However, the majority of participants were vaccinated with the second dose on 12 February 2021. The observed decline in seroprevalence is, therefore, not precise in time. Furthermore, the seroprevalence might be slightly underestimated due to the point-of-care test characteristics (sensitivity of 92.9%) or differ from the actual seroprevalence due to rapid test interpretation by different study investigators. Lastly, missingness completely at random (MCAR) in the outcome measure was assumed in this analysis. The time since self-reported COVID-19 infection prior to vaccination was not taken into account.

## 5. Conclusions

Nursing home residents are at high risk for developing severe cases of COVID-19 and were therefore prioritized in COVID-19 vaccine strategies. We found that the SARS-CoV-2 seroprevalence significantly decreased four and six months after vaccination among NHR, with infection-naive NHR, older NHR, or NHR with a high care-dependency level having significant lower chances of having detectable SARS-CoV-2 antibodies after vaccination. Among NHS, only significant decreases were observed after six months. These results demonstrate the need for close monitoring of the SARS-CoV-2 humoral immunity in this population, as NHR are prone to the rapid waning of SARS-CoV-2 antibodies upon COVID-19 vaccination. Although the majority of NHR in Belgium received a booster vaccine soon after the observations in the present study were made, follow-up studies are needed to assess whether SARS-CoV-2 antibodies are persistent upon booster vaccination.

## Figures and Tables

**Figure 1 viruses-14-02361-f001:**
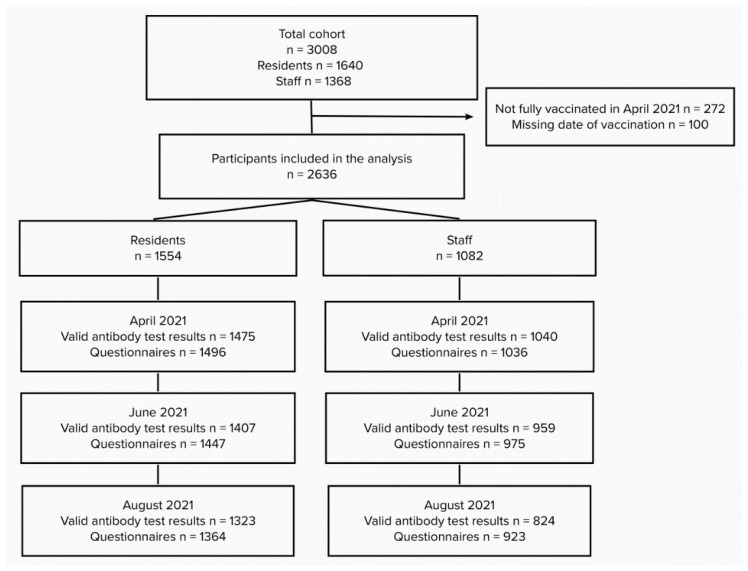
Overview of the total number of participants in the cohort and the participation in April, June and August 2021.

**Figure 2 viruses-14-02361-f002:**
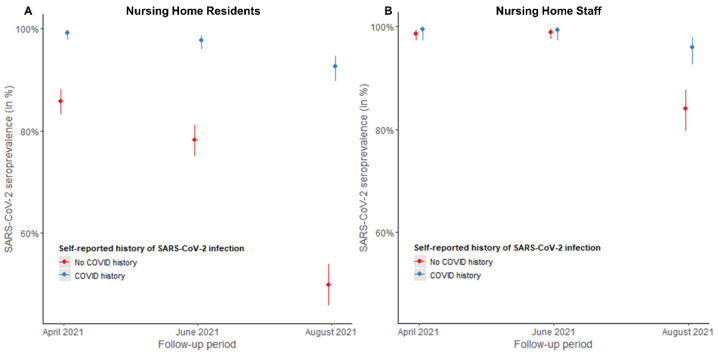
Adjusted seroprevalence among nursing home residents (**A**) and staff (**B**) with and without self-reported history of SARS-CoV-2 infection, Belgium, two (April 2021), four (June 2021) and six months (August 2021) after COVID-19 vaccination. SARS-CoV-2 seroprevalence was estimated based on a generalized estimating equation model averaged over nursing homes. Blue dots represent the seroprevalence among participants with a self-reported history of SARS-CoV-2 infection; red dots represent the seroprevalence among participants without self-reported history of SARS-CoV-2 infection. The error bars represent the 95% confidence intervals of these means. A history of COVID-19 infection was defined as a previously self-reported positive PCR and/or rapid antigen test, and/or CT-scan between February 2020 and the moment of antibody testing.

**Table 1 viruses-14-02361-t001:** Number and adjusted prevalence of SARS-CoV-2 antibodies among residents and staff in Belgian nursing homes per age group, two (April 2021), four (June 2021) and six months (August 2021) after COVID-19 vaccination.

	April 2021		June 2021		August 2021	
	Number ^a^ Positive/Total	Prevalence % (95% CI ^b^)	Number ^a^ Positive/Total	Prevalence % (95% CI ^b^)	Number ^a^ Positive/Total	Prevalence % (95% CI ^b^)
**Residents**						
60–69	102/107	95 (89–98)	99/102	97 (92–99)	68/98	68 (58–77)
70–79	216/227	95 (90–97)	200/218	91 (86–94)	151/211	70 (63–76)
80–89	647/721	90 (87–93)	588/692	85 (81–88)	437/648	68 (62–73)
90+	352/390	90 (87–93)	306/366	84 (79–88)	229/339	69 (63–74)
TOTAL	1345/1475	91 (89–93)	1219/1407	87 (83–89)	904/1323	68 (64–72)
**Staff ^c^**						
<30	171/172	99 (96–100)	152/153	99 (95–100)	128/139	92 (87–95)
30–39	267/269	99 (97–100)	234/237	99 (96–100)	174/197	88 (83–93)
40–49	270/272	99 (97–100)	253/255	99 (96–100)	181/203	89 (84–93)
50–59	293/299	98 (95–99)	286/289	99 (97–100)	233/265	88 (83–91)
TOTAL	1029/1040	99 (98–99)	950/959	99 (98–100)	734/824	89 (86–91)

^a^ Total number with valid test result, participants who did not complete the date of birth in the questionnaire are only included in the total number of residents, but not in the specific age categories. ^b^ CI, confidence interval. ^c^ 33 staff members were ≥60 years old, age category not shown.

**Table 2 viruses-14-02361-t002:** Association between SARS-CoV-2 seropositivity and age, self-reported history of infection, sex, comorbidities, care dependency and time since vaccination).

	Odds Ratios for SARS-CoV-2 Seropositivity					
	Residents ^a^			Staff ^b^		
Predictor	Odds Ratios	95% CI ^c^	*p*-Value	Odds Ratios	95% CI ^c^	*p*-Value
(Intercept) ^d^	8.18	6.18–10.82	<0.001	78.19	40.71–150.18	<0.001
4 months after vaccination (June 2021)	0.60	0.50–0.72	<0.001	1.34	0.66–2.70	0.417
6 months after vaccination (August 2021)	0.17	0.13–0.20	<0.001	0.08	0.04–0.15	<0.001
History of SARS-CoV-2 infection ^e^	19.43	7.73–48.83	<0.001	2.38	0.49–11.55	0.281
Male gender	0.84	0.64–1.09	0.193	0.70	0.40–1.22	0.212
Age	0.80	0.69–0.91	0.001	0.87	0.72–1.05	0.146
Presence of comorbidities: 1 or more ^g^	0.84	0.66–1.07	0.164	NA ^f^	NA ^f^	NA ^f^
High care dependency level: C, Cd or D ^h^	0.70	0.56–0.88	0.002	NA ^f^	NA ^f^	NA ^f^
(June 2021 * History of COVID-19 infection) ^i^	0.59	0.26–1.32	0.199	0.69	0.14–3.33	0.644
(August 202 1* History of COVID-19 infection) ^i^	0.64	0.26–1.57	0.329	1.89	0.38–9.38	0.435

^a^ Adjusted generalized estimating equation model for residents with time-by-self-reported infection for SARS-CoV-2 seropositivity with reference levels: April 2021 (baseline), no self-reported history of SARS-CoV-2 infection, no comorbidities, median age of 87 years old, female, care level O, A and B. ^b^ Adjusted generalized estimating equation model for staff with time-by-self-reported infection for SARS-CoV-2 seropositivity, with reference levels: April 2021 (baseline), no self-reported history of SARS-CoV-2 infection, no comorbidities, median age of NHS of 43 years old, female. ^c^ confidence interval. ^d^ The Odds for seropositivity for the reference profile. ^e^ History of infection is defined as a previously self-reported positive PCR-, -/antigen rapid test, and/or CT-scan between February 2020 and antibody testing. ^f^ NA, not available. ^g^ At least one of the following comorbidities (cardiovascular disease, diabetes, hypertension, immunosuppression, severe renal/lung/cardiac disease, active cancer). ^h^ As according to the ‘Katz Evaluation Scale’ (independency in activities of daily living). ^i^ Interaction terms.

## Data Availability

The datasets used and/or analysed during the current study are available from the corresponding author on reasonable request and with permission of Sciensano (Belgian Institute for Health).

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
