# Peer review of "Declining Prevalence of SARS-CoV-2 Antibodies among Vaccinated Nursing Home Residents and Staff Six Months after the Primary BNT162b2 Vaccination Campaign in Belgium: A Prospective Cohort Study"

_viruses, 2022, doi:10.3390/v14112361_

Round 1
Reviewer 1 Report
The manuscript is quite interesting in describing risk of SARS-CoV-2 infection after vaccination. I have 1 each major and minor comment respectively ;
1. Authors need to describe study population in Belgium concisely. Where 69 NHs comes from and how did you choose from total NHs?
2. In the analysis, authors describe intercept which is not well used in both result and discussion. This can be omitted.
Reviewer 2 Report
Declining prevalence of SARS-CoV-2 antibodies among vaccinated nursing home residents and staff six months after the primary BNT162b2 vaccination campaign in Belgium: a prospective cohort study
This is an interesting and relevant study by the authors for the current scenario of the COVID-19 vaccine status specifically emphasizing older and comorbid persons. The study is meticulously designed and well executed; results are presented clearly. The manuscript is scientifically sound and easy to follow. However, the following are specific comments to strengthen the present manuscript further.
Specific comment,
1. A history of COVID-19 infection was defined as a previously self-reported positive PCR and/or antigen rapid test and/or CT scan between February 2020 and the moment of antibody testing.
From this statement in figure 2, legend it is not clear whether the history of COVID-19 was before or after vaccination, there is a possibility that participants got infected with SARS-CoV-2 variants after the second dose of vaccination time before two, four, or six-month time point. Any specific reporting data of infection time point may further help in obtaining inference for this higher seroprevalence among infected NHRs compared to infection naïve NHRs.
2. Figure 2 warrants a bigger font size for better visualization.
3. The rapid waning of SARS-CoV-2 antibodies upon COVID-19 vaccination is a major concern for older persons and persons with comorbidities having immunosenescence. Hope is for the booster, does the booster will provide long-lasting immunity? Or we may consider follow-up studies or continue vaccination at a regular interval, any comment on this?
Author Response
Please see the attachement
